# Perspectives for the Development of a Circular Economy Model to Promote Ship Recycling Practices in the European Context: A Systemic Literature Review

Francesco Tola *, Enrico Maria Mosconi, Marco Marconi and Mattia Gianvincenzi

Department of Economics, Engineering, Society and Business Organization, Università degli Studi della Tuscia, 01100 Viterbo, Italy; enrico.mosconi@unitus.it (E.M.M.); marco.marconi@unitus.it (M.M.); mattia.gianvincenzi@unitus.it (M.G.)
* Correspondence: francesco.tola@unitus.it

**Abstract:** The recovery of materials and components from end-of-life (EoL) ships necessitates the comprehensive demolition of vessels and the implementation of reuse processes to achieve the "circularity" of raw materials, which has potential benefits for economic and environmental sustainability. The European Union's (EU) legislative policy, as reflected in the Ship Recycling Regulation, has been shifting towards the establishment of green markets for ship dismantling. Various literature reviews have focused on investigating EoL management and demolition activities for ships, as they play a crucial role in promoting sustainability in the ship value chain. This research aims to enhance the current state of knowledge by linking ship recycling and life cycle management activities with circular economy models. The goal is also to introduce a conceptual framework for the effective recirculation of components and raw materials. Scientific publications have been collected, reviewed, and categorized into strategic clusters to identify current and future challenges, to establish a path for potential developments in a circular economy model for ships, and to suggest future research directions that would support the implementation of a circular economy system for ship eco-design, life cycle management, and recycling. Findings provide valuable insights, particularly regarding the recognition of environmental benefits, business opportunities, and the opening of green markets in the context of ship recycling in the EU.

**Keywords:** circular economy; ship recycling; circular model for ships; systematic literature review

## 1. Introduction

The world is currently grappling with a predicament caused by the escalating accumulation of waste and inadequate waste management practices. The proliferation of waste, including both rural and urban types, causes harm to the planet and contaminates soil, water bodies, and the atmosphere to a worrying degree [1]. Over the past few decades, tangible efforts have brought about considerable changes in our wasteful behavior, leading to the availability of several alternatives for waste reduction. Among these options, the utilization of recycling has gained widespread acceptance as a successful and eco-friendly solution for waste reduction and as a green method of curbing environmental degradation [2]. Recycling falls under the main principles of circular economy (CE) thinking [3]. Transitioning from a linear economy to a CE requires enabling conditions that remove existing barriers to product expansion and material recovery possibilities. The concept of CE was developed to reverse unsustainable development models and create long-term prosperity [4]. In the CE paradigm, every economic activity should maximize the functioning of the ecosystem and human well-being [5].

As defined by the Ellen MacArthur Foundation [6], Circular Economy (CE) is a systemic approach to economic growth designed for the benefit of businesses, society, and the environment. It is a regenerative economic model which aims to gradually decouple

growth from resource consumption. CE seeks to implement a more efficient and sustainable economic system, reducing waste based on cradle-to-cradle thinking. CE can be considered a sustainable growth strategy to address the serious challenges of environmental pollution and resource depletion. The transition from a linear economy to a CE requires compliance with certain principles: (i) preserving natural capital and renewable resources; (ii) improving resource yields by recycling materials and ensuring that energy is produced from renewable sources; (iii) ensuring that all resources are used to generate value by reducing negative externalities [7]. In the literature on CE, some authors explore differences between the various closed-loop or circularity options, which are described as 3R, 4R, 5R, 9R, and even 10R principles [8]; however, the three Rs—"Reduce, Reuse, and Recycle"—are the fundamental principles. They are considered distinct waste management principles and constitute a consolidated notion of CE both in theory and in practice.

Therefore, the limits of ecological sustainability are expanded, and industrial outcomes are transformed to create a balanced connection between the natural world, economic development, and human well-being. Greater material circularity offers organizations multiple economic benefits, such as reduced material expenses, increased resource utilization, and heightened resilience [9], making a positive contribution to both the planet and the wider community [10].

Focusing on ships, their disposal at the end of their useful life creates substantial waste, representing a potential danger to the environment but also huge potential for material recovery [11]. Maritime transport represents the backbone of the global economy; in fact, almost 90% of the goods traded globally travel by ship, and it is physically impossible to replace this with another means of transport [12]. Today, there are nearly 100,000 ships, of which 1–2% reach the end of their life annually [13]. The average service life of a ship ranges from 20 to 30 years [14].

Ship recycling refers to the process of dismantling ships to extract and recover materials, particularly steel, which comprises 95% of the material used in their construction, and other common materials such as copper alloys, titanium and titanium alloys, aluminum and lead, and various pieces of electronic equipment that can be reused or recovered for their valuable materials [15]. It is estimated that recyclable materials account for 95–98% of the light displacement tonnage (LTD) of a ship [11]. The recycling of ships involves the technical procedure of disassembling outdated vessels to retrieve usable materials in a secure and eco-friendly manner [16].

While ship recycling offers significant benefits for the three pillars of sustainability, economy, society, and the environment, the current system faces some obstacles. Coastal regions of South Asia witness an industry whose environmental and health balance urgently needs change; such pressures are due to the techniques of dismantling and the standards of recycling plants not being compliant with international standards for worker safety and marine environmental protection [17]. Large amounts of carcinogenic and toxic substances (Polychlorinated biphenyl, PVC, polycyclic aromatic hydrocarbon, tributyltin, mercury, lead, isocyanates, sulfuric acid) are discharged into the soil and coastal waters, posing serious risks to the safety of workers.

The ILO has identified shipbreaking as a highly perilous occupation globally [18]. According to data released by the NGO Shipbreaking Platform, between 800 and 1100 ships arrive on the shores of South Asia for dismantling every year, using poor disassembly techniques. Countries such as India, Bangladesh, and Pakistan handle almost 90% of the global gross tonnage scrapped altogether. The attractiveness of the ship recycling and breaking business in these countries is determined by the high price offered by yards located in Southeast Asia for the purchase of ships, as well as by their weak environmental and labor safety regulations [19]. It is expected that the ship demolition industry will become more prosperous as the number of ships produced worldwide increases [20].

Given the evolving global socioeconomic landscape, establishing a sustainable sector is crucial for long-term viability. Therefore, a thorough study of modern dynamics is required to develop a practical and sustainable process that reflects the principles of CE, to estimate

reusable materials and waste while ensuring efficient standards and achieving total quality management (TQM) [21–23]. Europe wants to play an active role in this activity and wants to counter the export of ships flying the European flag to different areas of Southeast Asia. Europe's position in making ship recycling attractive and stimulating within its own yards is strongly linked to the European Green Deal and the Circular Economy Action Plan 2 [24]. Notwithstanding the key worldwide ship recycling players, the structures of South Asia, witnessing environmental damage, professional risks, and socioeconomic injustices, the sector is closely linked to the objective of wide-reaching sustainable progress as a resource utilization endeavor.

The study is arranged in this manner: the opening part addresses the research background with an overview of ship recycling's history and materials, details on disassembly methods and techniques, regulatory frameworks, and current studies on the implementation of circular economy models. In the second section, the research method is defined, including the research questions and how the selected articles were catalogued. In the third, the results are presented, and finally, the main results are discussed.

## 2. Ship Recycling Overview

A vessel has three distinct phases in its existence: (i) designing and building as asset creation, (ii) shipping operations as upkeep, and (iii) EoL dismantling. Vessels are typically taken out of service using methods such as ship scrapping, disarmament, abandonment, shipbreaking, and recycling. The latter is often viewed as the optimal method for disposing of a ship once it reaches the end of its functional life.

### 2.1. History of Ship Recycling

Ship recycling has been a long-standing tradition, as shipbreaking played a significant role in financing the industrial revolution [25]. During World War II, countries such as the US, the UK, and Germany produced a distinct category of ships, referred to as the "obsolete fleet." The retirement of these ships spawned the inception of "Obsolete Vessel Scrapping", a novel industry, which has developed in response to advancements in engineering and changes in global socioeconomic conditions. In the mid-20th century, the shipbreaking industry was mainly located in the ports where the obsolete fleet originated [26]. In the 1950s, ship demolition moved to the Spanish and Italian Mediterranean coasts and to Japan [27].

Before the 1960s, demolition was largely focused in developed nations such as the US, the UK, and Germany [28]. During the 1970s, ship demolition shifted towards emerging nations such as Spain, Turkey, and Taiwan, mainly because of the abundance of low-cost workforces and the existence of a thriving steel re-rolling industry. Beginning in the early 1980s, ship owners began sending their end-of-life ships to shipbreaking yards in India, Pakistan, Bangladesh, the Philippines, and Vietnam in order to maximize their earnings, despite the minimal health and safety and environmental (HSE) standards in these countries.

Currently, the main ship recycling destinations are in South Asia, with two key elements fueling this trend: the inexpensive workforce costs and the ships' selling profit, as the beaching method is economically more favorable [29]. Ship recycling provides the main source of livelihood for numerous people in South Asia. This industry provides the main source of steel [23] and significantly contributes to the local shipbuilding industry [30], but at the same time, it has a negative impact on the environment in the ship recycling areas.

Ref. [31] investigated over 22,500 business records of ships scrapped from 2000 to 2019, estimating that 22,547 ships were scrapped worldwide, representing 357,365,473 GT dismantled. 80% of the main ship recycling destinations are in South Asia; even though the European Union has a large ship demolition yard park, only a small portion of the ships sent to demolition yards reaches European coasts. As stated by the [19], the recycling capacity of facilities located in OECD countries is insufficient to meet the demand for ships intended for recycling, and thus the recycling rate is less than 5%.

### 2.2. Materials Composition

The amount of waste produced in a shipyard is dependent on factors such as the quantity, dimension, and type of vessels being recycled, as well as the recycling percentage, which can be influenced by the ship's material composition, the technology used, the demand for reusable or recyclable products, and relevant legislations [32]. As a result, it is challenging to precisely estimate the amount of waste generated by a specific ship recycling yard [33,34]. The dismantlement of ships can have severe consequences for both the environment and public health due to the release of toxic chemicals. These chemicals can include dangerous substances such as asbestos, polychlorinated biphenyls, lead, mercury, ozone depleting substances, residual fuels, and anti-fouling pesticides. Failure to properly handle the release of toxic chemicals during ship dismantling can result in a substantial hazard to surroundings [35]. Indeed, it is not uncommon to encounter instances where elevated levels of metal pollutants have been found in the sediment in the beach close to operative yards [36].

Determining the average output of waste and reusable materials from ships at the end of their lifecycle, regardless of type and size, presents a significant challenge. There are only a few studies available on this subject. The study conducted by [32] reviewed and discussed studies by [37–39]. However, no solutions have been found for recyclers to evaluate the makeup of materials in a vessel that has reached the end of its operational lifespan. The study by [32] introduced a method for determining the materials that comprise an EOL 11044 LDT, built in 2006, based on the stability manual. Nonetheless, the extent of their research was restricted to just one vessel.

Several authors in the [40] stated the need for adequate and efficient basic knowledge support for ship design, production, life cycle management and recycling activity worldwide [41]. More specifically, the document by the [40] categorically suggested that any recycling industry that wants to survive for a long period of time, especially one located in Europe, will have to orient its activities towards the implementation of an effective basic knowledge support mechanism. Ref. [42] gave a comprehensive examination of the context of data management and the creation of a suitable tool that satisfies the demands and necessities of a binding tool in ship recycling. Ref. [43] considered various factors involved in ship recycling and proposed an informative framework to support the decision-making process for the dismantling of obsolete ships. The proposed framework integrated a dynamic simulation tool with decision-supporting features to cater to the individual needs of the different stakeholders involved. Ref. [44] highlighted the point of industrial plant development as a process involving wide interaction between numerous layout design parameters.

The research emphasized some of the key phases of the procedures and proposed a concise yet valid methodology for modeling the vital components of recycling facilities. Ref. [45] deemed the design for disassembly (DfD) approach to be the best solution for a safe and sustainable recycling process. Ref. [46] presented a model for assessing the future market of ship demolition.

### 2.3. Ship Recycling Yards/Facilities

Various techniques are used for ship scrapping, to which correspond varying costs and degrees of environmental and social impacts. Main methods for ship scrapping and the associated recycling of materials have been analyzed by [47], who identified four methods that vary in terms of cost, technology, workplace safety, and environmental impact. These are the Beaching, Landing, Alongside, and Dry-docking methods.

In the Beaching method, mainly utilized in Bangladesh, India, and Pakistan, the ship has its cargo and ballast cleared out and is pushed towards high-tide sea level. Laborers then break it down into smaller sections, which are brought closer to shore. Once on land, everything is cut into progressively smaller pieces and eventually transported away by rubber transport means. This working method is very economical, but worker safety is widely ignored. Often, critical material leaks also occur, and the materials are released

into the surrounding environment, threatening not only the environment but also local communities [48]. Today this process is the most used one—almost 80% of EoL ships are scrapped with this method. India, Pakistan, and Bangladesh represent hot spots for ship scrapping due to the configuration of their coasts, which feature large tidal distances and wide stretches of mud [49].

The Landing technique is usually employed in regions where the tidal current is weak and easily forecasted. This kind of process facilitates the control and prevention of toxic substance leaks into the sea. The ship is collided with the seashore or with a concrete landing quay. A mobile crane working from the shore makes the first step of scrap-lightening the ship. Subsequently, operations of lifting, access, and dismantling that may also involve the use of temporary wharfs or semi-permanent piers [50] are carried out. This method resembles beaching but is comparatively safer. Currently, Turkey is the main proponent of the Landing method.

The technique called Alongside or Pier-breaking consists of a ship being dismantled with a crane, from the top to the hull, while it is blocked on a quay or pier. Typically, the final part is elevated for the purpose of completing the cutting process. This method is frequently used in China, the EU, and the United States. The impact of any pollutants is lesser compared to the two previous demolition methods, given that the prevention and monitoring of procedures can be contained by not releasing any spills into the sea.

In the Dry-docking method, the ship is brought into a dry dock that is both closed and flooded. Once the water has been drained, the ship is accurately disassembled. This approach offers a controlled setting for demolition work, reducing the threat of contamination and its overall impacts. Despite its benefits, the method is not commonly used due to the elevated costs involved in building and maintaining the dry dock. Although this method is considered the securest and most environmentally-friendly way to recycle ships, it is rarely employed due to its capital and maintenance costs. The Dry-docking method is practiced in Europe and the United States. The probability of accidentally polluting the surrounding waters is practically nil, given that everything is contained in the dry dock [51].

The substantial number of ships landing on the shores of South Asia is frequently tied to the most appealing purchasing offers for EoL ships typically coming from shipyards located on the Indian subcontinent. According to pricing data from [52], shipyards located in Bangladesh maintain the lead with typical prices of $610 per ton for the purchase of container ships, $600 for tankers, and $590 for bulk carriers. They are followed by shipyards in Pakistan, with approximately ten dollars less for each class, and then in India, with another twenty dollars less. Prices in Turkey have remained stable at respectively $300, $290, and $280. The economic surplus due to the sale of ships to substandard recycling centers has not incentivized many shipowners to engage in "green recycling", which is recycling by following international HSE regulations, shifting the cost to underprivileged communities [27].

### 2.4. International Efforts to Support the Safe Recycling of Ships

The persistent challenge of end-of-life ship management and ship recycling materials is a global issue. For some time, political actors have been striving to establish more efficient regulatory frameworks in order to stop vessels from ending up in subpar ship recycling yards.

### 2.4.1. The Basel Convention

The Basel Convention (BC) [53], which became effective in 1992, is an international treaty signed by 184 nations with the goal of reducing hazardous waste production and regulating its transboundary movement. This is particularly crucial for the ship recycling industry. The prior informed consent principle is at the heart of the system, meaning that before any shipment of hazardous waste, the receiving country must be notified and give written approval. In 2003, the Basel Convention issued technical guidelines for the safe and environmentally responsible demolition of ships. These guidelines cover all aspects of

the process, from the environmental management plan to the actual ship recycling facility, and offer recommendations on implementing best practices and monitoring performance. Implementing the provisions of the Basel Convention for ship recycling, however, has proven challenging, leading the Conference of Parties to request that the IMO create a binding legal instrument.

### 2.4.2. The Hong Kong Convention

In May 2009, the IMO adopted the Hong Kong International Convention for the Safe and Environmentally Sound Recycling of Ships (HKC) [54], which has been ratified by 63 nations. The purpose of the HKC is to prevent end-of-life ships from posing hazards to human health and safety or the environment. The convention covers all aspects of a ship's life cycle, including design, construction, operation, and safe recycling, and governs the operation of ship recycling facilities. The HKC addresses hazardous materials, such as asbestos, heavy metals, and others, present in ships sold for scrap and environmental concerns related to ship recycling methods and conditions. The convention provides guidelines on ship recycling plans [55], facility authorization [56], ship inspection [57], and the control and certification of ships [58]. The HKC requires ships to carry documentation of hazardous materials and to be periodically inspected until recycled. The convention only applies to ships of 500 GT or more, regulates the recycling status of ships, owners, and flag states, and will come into effect 24 months after being ratified by at least 15 countries with 40% of the global fleet and 3% of their recycling capacities.

### 2.4.3. European Regulations on Ship Recycling

The handling of retired ships in the EU has changed from the waste regulation (Regulation 1013/2006) to a separate system that heavily relies on the jurisdiction of the ship's flag state (Regulation 1257/2013 on Ship Recycling). It is important to mention that the EU is a member of the Basel Convention. Regulation 1013/2006 aims to follow not only the Basel Convention but also the OECD Decision and the Lomé IV Convention. In preparation for the implementation of the Hong Kong Convention, in 2013, the EU established the EU Regulation on Ship Recycling (EU SRR), which came into effect on 30 December 2013. As of 31 December 2018, ships with EU flags over 500 GT and ships visiting EU ports must be recycled in safe and eco-friendly facilities, per Article 2 of the Regulation. The EU SRR sets higher standards than the Hong Kong Convention, such as a ban on the Beaching method, the safe handling of toxic waste, and the protection of worker rights. Article 19 mentions a potential financial instrument to prevent evasion of the European list of approved ship recycling facilities. Article 8 mandates that new ships with contracts signed after 31 December 2018 must have a hazardous materials inventory, and ships with EU flags or visiting EU ports must have a certified inventory after 31 December 2020. The European Commission, according to Article 16 of the EU SRR, updates a list of approved recycling centers in the EU and beyond. The list was created on 19 December 2016 and is updated regularly to add or remove facilities that meet or fail to meet the standards. The latest Commission Implementing Decision (EU) 2022/691 on 28 April 2022 recognized 44 facilities in the EU and nine outside the EU as compliant. There are no compliant facilities in Southeast Asia on the list. Figure 1 shows the annual recycling capacity of European yards.

The various attempts by political actors to introduce a globalized California effect in the ship recycling industry have not produced the desired environmental and social standards. Circumventing international law through the practice of flag of convenience is a common practice.

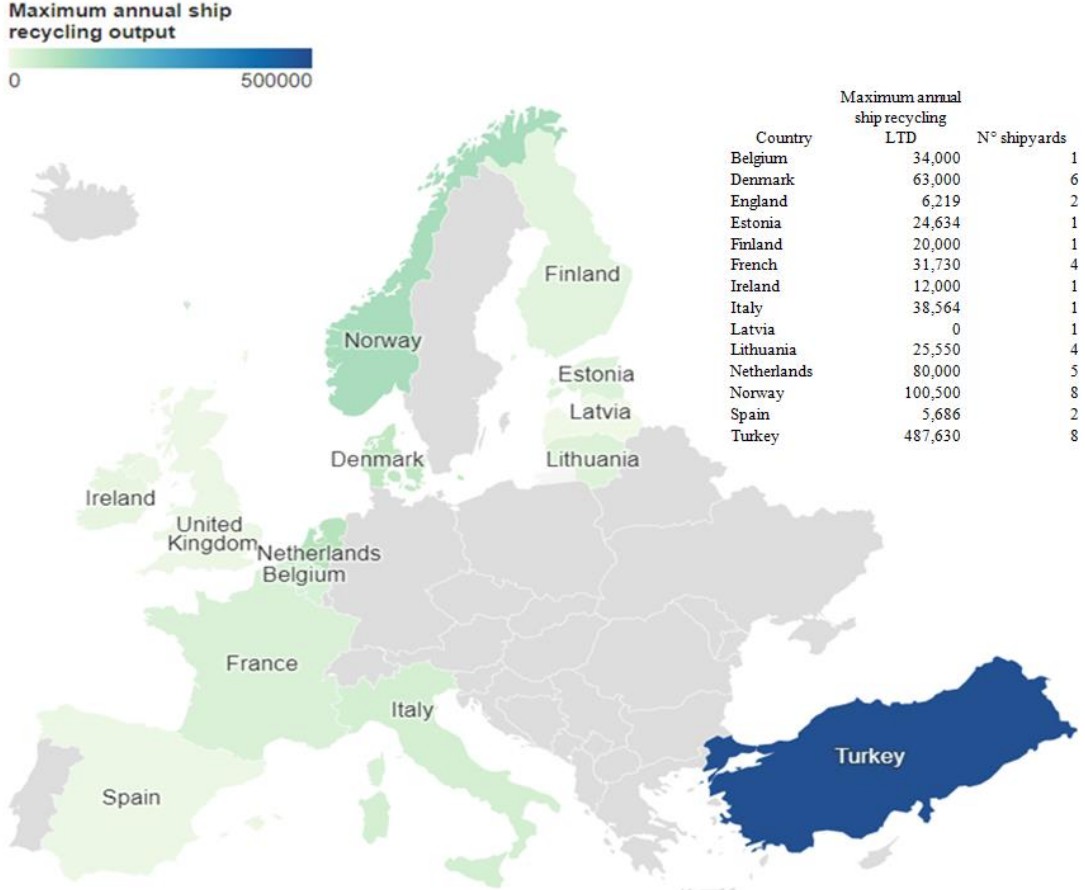

**Figure 1.** Recycling capacity in LTD in European countries. Commission Implementing Decision (EU) 2022/691 of 28 April 2022. Source: authors' elaboration from (EU) 2022/691 of 28 April 2022.

*2.5. Circular Economy as a Lever for Ship Recycling*

The potential of CE in the ship recycling sector can allow for the reduction and improvement of material flow management control, and it also reduces the need for virgin material input in production processes and thus helps to close the material flow cycle [59]. It is expected that the ship demolition industry will thrive as the number of ships produced worldwide constantly increases [20]. Circular Economy (CE) for ship recycling represents significant potential as most of the materials extracted from ships can be recovered to be used as secondary raw materials. The economic value of a ship at its end-of-life can be the source for creating green markets, which encourage the development of market opportunities for the supply of critical raw materials to improve upstream design and downstream recovery for new industrial processes. With reference to the latent potential of CE in the ship recycling sector, this article investigates the opportunity, feasibility, potential effects, and limitations for the implementation of a CE model for ship recycling to improve knowledge on the subject and provide a framework that can be used for future research. The implementation of CE business models can involve enormous benefits as demonstrated by [60].

By utilizing technological and organizational innovations that are socially, environmentally, and economically beneficial, businesses can better achieve their social and environmental goals. By adopting environmentally, socially, and commercially advantageous technology and solutions, organizations can achieve their social and environmental goals by innovating their business models. The potential application of a CE model in the ship recycling industry could, on the one hand, support the renewal of the world's ship fleet and maintain a balance between supply and demand; on the other hand, the recycling of

millions of tons of waste materials can support the realization of a circular flow for resource supply. The scope of the study stems from the EU's strategic ambition to transition to a low-carbon, resource-efficient economy and to support growth and employment and make the sector in question competitive. The research questions are as follows:

**RQ1.** *Is it feasible to implement a circular business model for ship recycling in Europe?*

**RQ2.** *Is it feasible to implement a CE strategy for ship recycling in Europe?*

The topic of ship recycling has previously been discussed in the scientific literature's analysis section; Ref. [61] conducted a literature analysis to summarize the main investigations and arguments leading to sustainable development. Ref. [59] discussed the theoretical and managerial implications on ship recycling companies and policy makers. Ref. [34] have examined the research on shipyard recycling procedures to find and validate sustainable methods in the ship recycling supply chain for recommendations on how to adopt best practices. Ref. [62] have examined the literature on existing ship dismantling techniques, compiled a list of hazardous emissions from decommissioned ships, analyzed their movement in various environmental sectors, and offered suggestions for the distribution of responsibility.

Many literature evaluations concur that interdimensional research is necessary to create regulations that promote circularity and management tools. Our research on ship recycling aims to close several gaps in the body of knowledge. As a matter of fact, the majority of systematic literature evaluations that are now accessible are primarily concerned with the environmental and social sustainability of the ship destruction process, ignoring the significance of the circular economy in this context. Furthermore, many earlier studies focused only on a few well-described geographic locations, ignoring the examination of additional crucial circumstances for the ship recycling industry. The examination of the risks associated with ship disassembly is the topic of several reviews in the literature. This study aims to further understanding of the ship demolition process and its effects on the circular economy, as well as sustainability, by presenting additional research avenues and providing practical resources for the development of successful European policies and initiatives.

## 3. Methodology

The article is a systematic analysis of the literature on the end-of-life management of ships, with a focus on economic feasibility, environmental impact, social factors, technology, and regulation. The goal is to provide an overview for future research on ship recycling and develop a CE model. A systematic literature review was conducted to identify the main academic contributions on ship demolition. The study aims to find a solution to address the adversities in ship recycling through a CE-compliant recycling methodology. To answer the research questions, scientific studies on recycling activity were reviewed both in academic literature and in industrial practice. The literature review approach, as suggested by [63], consists of a research process based on the selection of academic databases, types of literature, and delimitations, choosing search terms, and a practical reading/screening within a specific period. Additionally, based on determining definitions and key concepts for analysis, all collected material is summarized to obtain final outputs and original findings useful for future perspectives. In particular, the first phase of the literature review process involved a systematic search to access published studies related to ship recycling activities.

### 3.1. Systematic Literature Review

The SLR technique discovers significant contributions pertinent to a particular research subject through a methodical, transparent, and replicable methodology [64]. In this instance, the goal was to examine the current status of research on ship recycling in order to gauge the supply chain's transition to CE. As suggested by [63,65], the SLR technique includes four main phases: (i) source identification; (ii) source selection; (iii) source evaluation; and

(iv) data analysis. The four phases of the review are illustrated in Figure 2 and discussed in the following sections. The PRISMA Checklist for this systematic literature review are available in the Supplementary Materials.

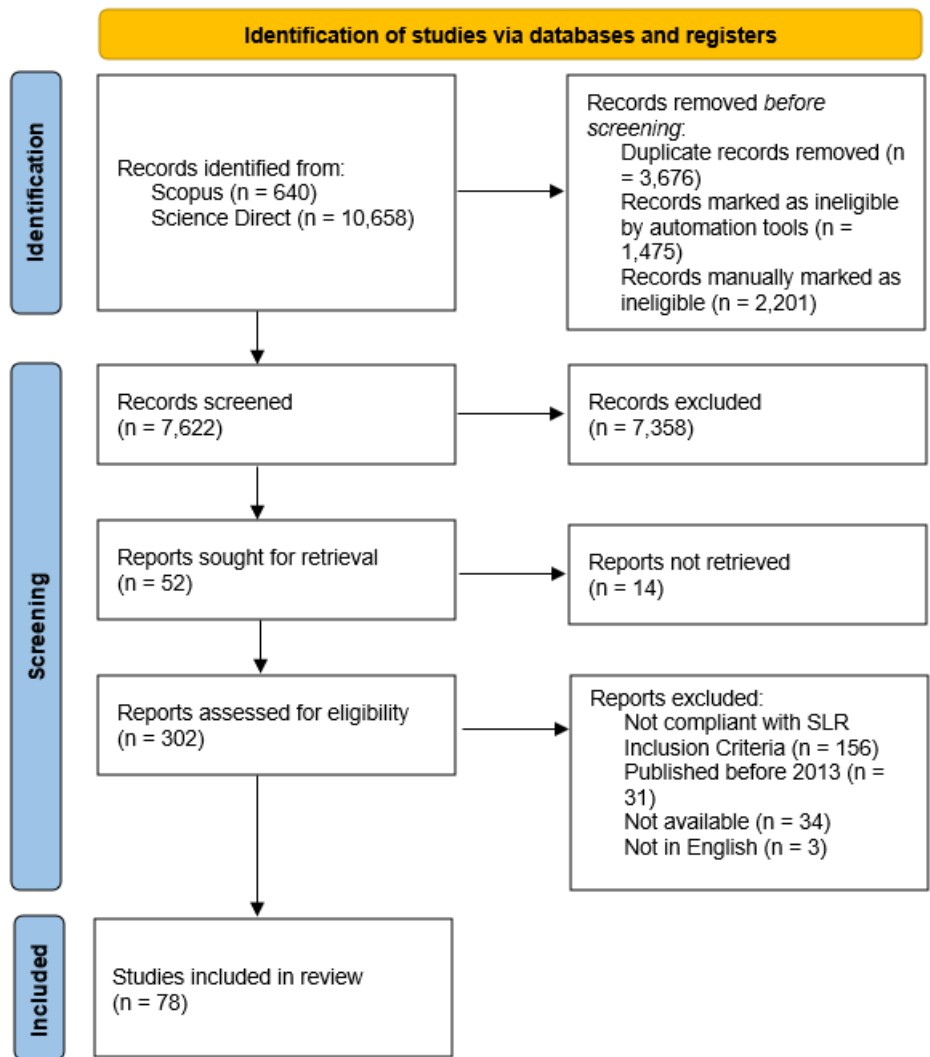

**Figure 2.** Papers search and evaluation process. Source: authors' elaboration.

### *3.2. Source Identification*

The peer-reviewed academic databases SCOPUS and ScienceDirect were used during the source identification step. The selection procedure was made more rigorous by using two sources simultaneously. To increase the number of articles included in the analysis, keywords were carefully selected. The following keyword string was used:

((ship OR vessel OR floating OR shipyard) AND (break OR scrap OR recycle OR green AND recycling OR dismantle OR demolition OR eol OR end-of-life OR decommissioning) AND (environment OR social OR economic OR technology OR regulation OR law))

A manual cross-checking process was conducted, and the Mendeley bibliographic citation software was used with the sort by title method to uncover and eliminate duplicate results.

### *3.3. Source Selection*

Once the subset of potentially relevant articles was identified, a first screening process of the abstracts was performed. To outline the boundaries of the analysis, the following inclusion/exclusion criteria were applied:

- Only articles in English were included.
- Scientific articles or industrial reports were included, while book chapters and conference proceedings were excluded.
- Articles published before 2014 were excluded.
- Publications that did not address the research subject (offshore renewable energy; marine biology; marine waste; navigation; piracy; international trade; ship modeling practices; ballast water treatment; waste management during navigation; ship stability; navigation computer systems; fishing; aquaculture) were excluded.

This scanning process resulted in a significant reduction of the number of papers (from 7622 to 302). This phase was handled separately by at least two members of the research group. During this, the choices taken to ensure an optimization of the selection process were evaluated. Studies that could not be completely rejected were included for further analysis and reading throughout the source evaluation step.

*3.4. Source Evaluation*

The resulting 302 papers were assessed and categorized according to their relevance in light of the requirements of the research topic. More specifically:

- Studies that develop possible sustainable development methodologies and possible CE actions for ship recycling were included.
- Studies that focus on the enhancement of shipyards for ship demolition were included.
- Studies that have contributed to the sustainable development literature or possible CE applications in the field were excluded.

This phase was entrusted to two members of the group, who operated independently, assigning each article to a possible reference cluster:

Economic cluster: The studies included in the economic cluster have analyzed the ship recycling sector from the perspective of economic analysis, demand, and supply forecasting of end-of-life (EOL) ships, economic feasibility, and financial tools.

Environmental cluster: Studies categorized under the environmental cluster have analyzed methods for environmental impact assessment, material classification, and more eco-friendly strategies for mitigating negative environmental externalities.

Social Cluster: The studies included in the social cluster have promoted a shared culture of responsibility, contributing to the protection of the environment and ensuring the health and safety of the workers involved.

Technological Cluster: The articles catalogued in the technological cluster have considered new technologies implemented in yards, industrial processes for the quantification and valorization of materials from EoL ships, and the design/planning of recycling activities.

Regulatory Cluster: The studies included in the regulatory cluster can be identified as the best practices and the strictest criteria to ensure that ship recycling is conducted in a sustainable manner, as well as the political decisions and strategies of the various political actors to improve the sector.

The final evaluation of the assignment of articles to reference clusters was carried out through a meeting among all authors. This approach ensured criticality in selection and inclusion in the clusters.

*3.5. Data Analysis*

The data were then imported into the Excel program, and a critical examination of the 78 papers that had been chosen was carried out with the intention of emphasizing essential activities and summarizing pertinent findings. In order to provide a thorough response to the study issues taken into consideration, the existing tools, methods, approaches, opinions, and strategies on ship recycling were surveyed.

## 4. Results

This study offers a thorough analysis of the literature with the goal of compiling a final list of scientific papers divided into five categories. The key findings of the examination

of the sample of articles are presented in this section. The analysis sample consists of 78 articles from various sources. Due to the interdisciplinary nature of ship recycling, the publications are from several study fields. The most frequent journals that deal with the topic of ship recycling are listed in Table 1.

**Table 1.** Top 5 most frequently publishing journals Source: authors' elaboration.

| No. of Publications | Journal |
|---|---|
| 17 | Journal of Cleaner Production |
| 12 | Marine Pollution Bulletin |
| 10 | Marine Policy |
| 9 | Ocean Engineering |
| 7 | Resources, Conservation and Recycling |

The publications range from 2014 to 2022. From 2019 to 2022, there was a sustained growth in the number of articles published (Figure 3).

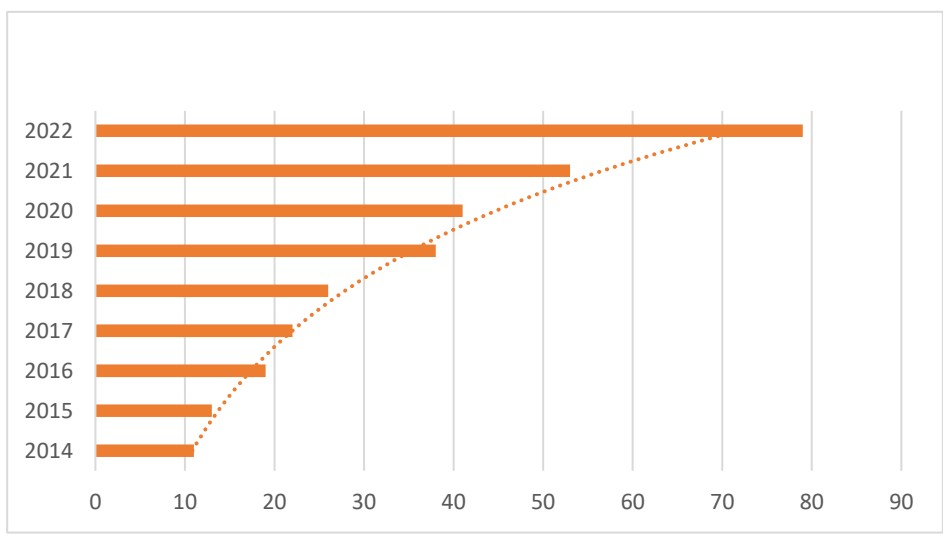

**Figure 3.** Ship recycling scientific articles by publication year Source: authors' elaboration.

The review provides insights for the development of the ship recycling industry, touching on economic, environmental, and social issues through implementing innovative technologies and regulations.

*4.1. Economic Cluster*

Ref. [20] established a dynamic material flow analysis (MFA) model for the world's merchant fleets, and a thorough examination of steel stocks and flows was carried out, demonstrating that reducing the number of EoL ships has a relatively modest impact on lengthening ship lifespans. They also confirm that there is a direct relationship between GDP and the number of ships sent for demolition. They also confirm the existence of a strong correlation between Gross Domestic Product (GDP) and ships sent for demolition. This supports the findings of [66] which indicate a negative correlation between economic growth and scrap ships. Economic assessments frequently neglect the need for steel as a significant downstream business generating demand for EoL ships. Analysis of material stocks can be a valuable solution to reduce the use of virgin materials by recycling and renewing ships. Ref. [67] analyzed the economics of demand and steel prices, concluding that South Asian shipbreakers can offer the highest prices for such ships due to high domestic steel scrap prices driven by population growth and urbanization in these nations. While industrialized countries' population growth is slower, their manufacturing sectors are more developed, and their urban–rural disparities are stable. To plan a ship demolition

market, it is necessary to understand steel consumption patterns, as a country's domestic steel demand is an attractive factor for demolition and material demand decisions [68]. Using a cost–benefit analysis approach to the ship's lifecycle, which estimates the net economic gain from recycling or reefing ships, is one potential way to promote the financial advantages of end-of-life ship management alternatives. This method can have positive effects on worker safety and environmental protection standards [47]. A survey directed at stakeholders can be an excellent knowledge-gathering tool that ensures valuable conditions for the development and expansion of industrial networks [69]. Ref. [70] suggested that to develop an excellent economic feasibility study for potential new players in the ship recycling sector, the questionnaire method is a good starting point. The questionnaire method can also be used for a successful marketing strategy. Ref. [71] aimed to define risks in the ship recycling industry using the Task-based learning approach, developing a conceptual model based on risk, stating that green management is the key point that defines the winning marketing strategy, and that reducing the time to market is essential as an important dimension of a responsiveness strategy to improve the trend in ship recycling yards in Europe and improve the environmental and social conditions of ship recycling yards in Southeast Asia. Ref. [72] proposed the use of financial tools to close gaps, using instruments that can guarantee funds or function similarly to a deposit system. This instrument can promote cooperation between shipyards, but it must work in conjunction with other regulatory instruments, which can be made easier by enacting a number of tax exemptions and deductions. Such subsidies can influence the supply of ships and the composition of the fleet. Ref. [73] examined the impact of subsidies on ship fleet renewal programs; the proposed model allows for fleet renewal by opting for an early demolition to return materials. The possibility of establishing a recycling license as sought by [74] may be economically feasible, provided that such adjustments, such as the use of incentives, are made, otherwise it would be classified as a hidden tax. To have a fair financial distribution and a fair decision-making process, for the scrapping of the ship, Ref. [75] believed that BCR and VAN can be used as financial performance indicators to determine the optimal EoL strategy. Ref. [76] noted that certain ship factors (such size and age) and market variables, in addition to the circumstances of the domestic market, affect the likelihood of a ship being demolished (charter rates and volatility, bunker prices, interest rates, and scrap prices), as confirmed by the studies of [77]. At the same time, Ref. [78] added that additional variables must be incorporated to capture market trends; the study argues that a crisis, such as the COVID-19 pandemic, could have an uncertain effect on the sector, given that market trading can increase the number of inactive ships. The utilization of indexes sets during the design phase of ships would facilitate the dismantling of the ship and the environmental risks associated with ship demolition [79]. Efficient communication regarding the availability and quality of scrap material results in increased demand and the immediate capacity for material regeneration [80]. Furthermore, improved communication and information within shipyards is necessary to increase material regeneration by designing an optimal closed-loop supply chain network with integrated direct and reverse logistics to examine the feasibility of ship recycling and regeneration [81]. A high level of collaboration amongst stakeholders, including ship owners, ship recycling businesses, and government agencies, is required to maintain a viable ship recycling industry [59].

The economic cluster demonstrates how methods and analysis that support the circular usage of recovered materials can be used to realize a circular economy business model. The major defense is that in order to accomplish this goal, a thorough market study must be done, taking into account not only the ships that are about to be demolished but also the quantity and caliber of the materials that can be salvaged. In order to better understand the need for circular materials, it is also critical to expand the collection of data, the processing of statistics, tight collaboration among supply chain actors for the purpose of data and information gathering, and the development of ad hoc indicators. Additionally, improved communication between shipyards and shipowners can help to create a supply chain for circular materials, which in turn can support the renewal of the ship fleet and maintain

a balance between supply and demand for ships. Increased communication between the parties can only be achieved with the use of incentives or other financing sources.

### 4.2. Environmental Cluster

The main cause of the current end-of-life management problems is a dearth of ecological components in the design and production phases [82]. Ref. [83] applies a life cycle assessment of ship demolition, suggesting that replacing virgin steel with recycled vessel scrap yields significant reductions in impacts. confirming the study by [84]. Additionally, the monitoring of materials and changing ship cutting techniques can significantly reduce environmental impacts in Southeast Asia. The LCA method proved to be a valid method for assessing the impact of the life-cycle of ship demolition and can guide shipbuilding and demolition yards towards designing green ships that comply with international environmental protection conventions [85]. The value creation shown in the LCA study provided by [86] demonstrated an allocation of value per environmental impact throughout a ship's life-cycle, with the peak reached in the ship's operation and the majority of the economic benefit being obtained during the use phase and remaining in Europe. They also suggest that choosing more robust environmental indicators can change impact assessments. Ref. [87], using the LCA method, stated the need for a standardized and comprehensive approach in measuring the environmental performance of shipbuilding activities and highlighted the difficulty of comparisons through the current LCA-based analysis methods. Studies on material quantification agree that the documentation and mapping of waste flows in terms of the number of wastes, divided by waste fractions, even before the ship arrives at recycling yards can improve the management and control of pollutants in air, soil, and water [23,38,62]. According to [88], the utilization of Material Flow Analysis (MFA), a tool commonly used in environmental engineering, has the potential to enhance ship recycling and the management of waste in ship recycling facilities. MFA identifies opportunities for improvement and ensures efficient use of resources, resulting in increased income. The study, through the MFA method conducted by [89], argued that the decision to correct recycling geography is a complex social, economic, and environmental phenomenon and must be handled with care, through information and technological systems that improve the ability to predict risk and waste management. The proposed computerized model by [90] can improve the choice of demolition process, controlling leaks and the spread of toxic substances in soil.

Most of the research included in the environmental cluster focused on well-defined regions such as the Asian South-east or Turkey and mostly explored the cycle of life inside the ships while they were being demolished. These studies frequently use a cradle-to-grave perspective. However, there are very few LCA studies on this topic in Europe, and there is no comprehensive comparison of the various recycling methods employed in shipyards. It would be acceptable to use more reliable environmental indicators not only during the demolition phase but also throughout the recycling and recovery of materials in order to improve research in this area. Moreover, specialized equipment is required for the management and containment of the risk of hazardous substance spills during ship breakdown and recycling. By examining the whole life cycle of ships and providing a more thorough analysis of the many recycling procedures used in shipyards all over the world, not only in specific locations, it is crucial to extend the focus of environmental research on the topic. The environmental effects and hazards associated with the destruction of the ships may be more fully understood in this fashion, and more effective efforts to mitigate them could be developed.

### 4.3. Social Cluster

Ship recycling in Europe is a volatile, low-frequency activity that depends on the release of ships in the scrap market. Ref. [91] stated that in European yards, even the work of cutting metals in the context of ship recycling fails to attract local labor and instead relies on migrant workers. The establishment of industrial networks would entail enormous

social benefits by promoting stable social conditions within a community [92]. Proceeding through a step-by-step methodology for risk management, implementable in any ship recycling yard in the world, could be a tool for dialogue for the "analysis team" composed of production managers, safety managers, safety supervisors, and the designated expert monitor, and serve as a systematic method for reducing risk to improve health and safety in shipyards [93]. In the ship recycling industry, worker safety is a common concern among governments, ship recycling companies, and political leaders worldwide. The results of the study conducted by [94] suggest that ship recycling businesses and political leaders can promote a strong safety culture and arrange regular safety training sessions; it has been suggested that operational dismantling safety directly determines worker safety, while hazardous materials, ship recycling management, and ship recycling equipment, as well as worker safety awareness, indirectly determine worker safety. Long-term exposure to asbestos for workers can lead to mesothelioma. Ref. [95] suggested that specific training programs to promote awareness of the proper handling of asbestos-containing products and more stringent monitoring of worker health and safety could decrease the number of deaths from this disease. Additionally, it is advised that both current and past employees have access to quality medical care. Ref. [15] investigated how HSE-related factors, recycling based on the size of the engaged yard, the ship's LDT weight, and the volatility of steel prices in the global market can justify the number of workers employed and the number of days required for the complete recycling of a given ship; as a result, it is possible to estimate the man-days required and subsequently hire teams for the phases and sub-phases of recycling and estimate completing the recycling activity within a certain time frame.

The social cluster's articles primarily focus on evaluating working conditions and morals in South-East Asian yards. Particular attention is paid to worker health and safety, taking into account issues such prolonged exposure to asbestos and other dangerous compounds, as well as inadequate training and working groups. The social sphere should be included in conventional life cycle assessment (LCA) techniques, along with pertinent indicators that are frequently not measured. There is now limited knowledge on the management and organization of work in Europe, which places it in a marginal position. Training programs and industrial networks could be established to expand the workforce in Europe in order to address this issue. This might enhance the working environment and encourage higher standards of morality and safety in South-East Asian shipyards.

*4.4. Technological Cluster*

The articles catalogued in the technological cluster can potentially play a constructive role in improving ship recycling activities; the studies have considered new technologies implemented in yards, industrial processes for the quantification and valorization of materials from EoL ships, and the design/planning of recycling activities. The guidelines suggested by [11], using HKC procedures, include creating a plan for ship recycling facilities and a ship recycling plan, using the idea of design for recycling when building new ships, optimizing the ship recycling process using material flow analysis (MFA), using better recycling equipment to increase recycling efficiency and reduce emissions and waste production, and using consolidate for producing goods in accordance with consumer demand. They also propose that the goal of increasing the revenue of the ship recycling process can be achieved by installing a waste-to-energy plant in ship recycling yards in order to generate additional revenue through the sale of waste-derived goods, and the goal of decreasing the costs of the ship recycling process can be accomplished by utilizing a design-for-recycling-based strategy. To promote the acceleration and adoption of a strategy based on design-for-recycling, the ship owner plays a key role in this stage, as they identify the technical and decision-making parameters of the ship's life-cycle. The ship owner allows for obtaining a ship that meets the technical assumptions of recycling, as well as for achieving high quality in the ships built by the shipyard [96]. To identify the difficulties related to end-of-life marine management, Ref. [97] suggest making it easier to incorporate sustainable metrics during the design process. A multi-criteria evaluation index

that conceptualizes a design hierarchy has been presented; it intends to support industry participants in the management of end-of-life maritime structures. Ref. [98] proposed a benchmark model, useful as a tool for decision-making strategies and allowing the designer to choose between various solutions in the early stages of the design process to reduce the impacts and environmental costs of the life-cycle and to standardize the LCI by identifying the necessary data and providing the necessary mathematical relationships; the software currently in use presents customization issues based on business needs. Ref. [32] used a new tool for quantifying materials on board ships to facilitate the ship dismantling phase through the ship stability manual and the WBS classification system. By installing a plasma gasification machine in recycling yards, waste may be transformed into valuable goods, generating income and ensuring recycling that is environmentally benign [99]. Possible solutions for mitigating impacts suggested by [100] include using water jet cutting to break apart ship sections; this can result in advantages in terms of economy, safety, efficiency, and environmental protection, as abrasive water jet cutting is a green technology and will not produce pollutants or dangerous materials caused by high temperatures. Ref. [101] stated that the use of water jet cutting can generate more waste produced by paint, which is rich in hydrocarbons, and suggest the use of thermal treatment such as pyrolysis as a good option for energy recovery or the recycling of carbon-related products, since they are promising approaches for the recovery of metal/metal compounds (such as zinc, copper, and titanium) from paint waste. While hazardous waste will be managed to prevent process contamination, the yard's design guarantees an environmentally friendly recycling process [102]. Ref. [103] proposed that systematic layout planning (SLP), one of the techniques for the effective design of ship recycling yards, can be optimized in order to overcome problems in recycling yards.

It is conceivable to apply and reproduce the technologies now used in ship demolitions, opening the door for a potential countertrend to the issue. Researchers have discovered technical ways to reduce environmental effects, including risk management, hazardous material conversion, and leakage monitoring, which could have a positive economic impact. It would be acceptable to adopt these technologies in other shipyards worldwide and assess their economic and strategic viability, taking into account each nation's full recovery chain, in order to boost their efficacy and reproducibility in crucial settings. The whole ship sector supply chain, including ship builders, ship owners, shipyards, and local disposal and recovery businesses, must be involved in the implementation of these circular economy methods.

### 4.5. Regulatory Cluster

The attention to law aspects represents a positive milestone towards making the ship recycling industry more circular but strongly depends on the regulations and standards that can be developed under national law; the law's objective can only be achieved if the regulations and standards developed under the law are environmentally respectful and have a holistic approach [104]. The primary causes of the negative effects of ship demolition activities, which prevent the industry's sustainable growth, are inadequate planning, management, legislation, and standards [105]. Overall, gaps in the BC's application to ships have not been filled internationally, and since it will be several years before the HKC enters into force, the maritime industry will have a fragmented regulatory framework until the conflict between the waste regime and ship recycling is resolved [48]. South Asia still lacks the institutional capacity to apply standards comparable to those of the EU. A greater emphasis on compliance with the law necessitates a review of several provisions of international agreements and domestic legislation because more ships are anticipated to arrive for recycling [106]. Ref. [67] proposed policy maneuvers led by Western states to increase the industrial standards of South Asian shipyards; an aid-based approach that best fits the polluter pays principle should be promoted. To avoid evading international law through the flag of convenience method, Ref. [31] argued that maritime authorities worldwide should consider digitizing paper records (already strongly required by the

European Union) and open data to the public. As laws become more stringent over the next years, more registers may open up as a result of the money they make from registering ships that will be destroyed for scrap. The implementation of international treaties must therefore be subject to stronger oversight, and this oversight should be predicated on a return to the idea of an actual link between the flag state and the ship [107]. The study conducted by [108] provided a current overview of European policies on ship recycling and supported an analysis that the application of the HKC at the European level is favorable and possible, reversing the problem of inadequate ship recycling that can only be effectively addressed through the early enforcement of the HKC and its application. A recent example of a public–private governance arrangement in ship recycling regulation is the recommendations proposed by [109], which improved the development of the regulation as follows: Additional recycling yards should be added to the European list, the application is taken more seriously, ship owners are exposed to greater responsibility concerns under civil liability law, and using the listed recycling facilities should become more alluring if the EU offers financial incentives. The public financial incentive study by [110] is based on an incentive/disincentive system: when the ship demolition industry booms, the government can collect more taxes. During recessions in the ship demolition sector, the government can grant the demolition industry some subsidies to maintain the normal operational costs of the sector. According to [111], financial instruments should be used to cover ship recycling insurance, port fees, and licensing to maintain the certification process on track and at the level of quality anticipated to prevent significant market distortions; as confirmed by the studies of [112], a framework for creating capacity will guarantee the complete compliance of recycling states through financial aid and knowledge exchange. Ref. [113] described the importance of adopting emerging measures for recreational ships, as they are excluded from the HKC and the SRR given that they do not reach the GT established to fall within the legal dictates.

The articles in the regulatory cluster examined ship recycling legislation and mostly assessed their advantages and disadvantages. In addition, they proposed legislative frameworks that would largely be applicable at the regional or national level. A precise and thorough definition would make it easier to enact more rigorous legislation, increase stakeholder collaboration, and streamline the recycling process. Furthermore, it is crucial to make sure that the rules and guidelines created in accordance with the law respect the environment and adopt a holistic approach that considers a ship's full life-cycle. To encourage information transmission and coordination in the recycling process, active participation from interested parties, including politicians, shipowners, and recycling facilities, is required. This will ensure the proper management of hazardous materials during the recycling process and help to improve compliance with regulations. In order to make sure that the industry runs in a sustainable and environmentally responsible manner, top-down political and regulatory frameworks need be put in place. To preserve quality and avoid market distortions, financial tools including insurance, port fees, and licensing, as well as certification procedures, might be used.

## 5. Discussion

The CE incorporates the three spheres of sustainability (economy, environment, and society); this can be achieved through the integration of technologies and the push of political actors [114]. Even though the topics have been chosen in a monothematic way, there is a strong connection between the clusters examined, so this systematic literature review's integrative goal is to synthesize the investigations and key topics in this field with the aim of reporting and reflecting on the trends and themes of the existing literature for all clusters under observation. The literature has been reviewed to produce a profound reflection for the realization of a CE model and achieve a secondary raw materials market for the ship demolition sector.

The key variables for understanding the ship recycling sector were identified during the review process. The articles were condensed into relevant keywords for the cluster.

Careful consideration of significance and relevance was needed when selecting these variables. To advance circular economy development, connections between variables have been established. The choice and inclusion of significant variables is dependent on the objective of the analysis, but it is crucial for improving the sector.

Key variables that can move the sector towards a CE system have been identified. The key variables and the interconnections between clusters are described in Table 2.

**Table 2.** Key variables and interconnection between clusters Source: authors' elaboration.

| Source | Variable | *Economic* | *Enviroment* | *Social* | *Technology* | *Regulation* |
|---|---|:---:|:---:|:---:|:---:|:---:|
| [20,47,66–68,80] | Market Analysis | ✓ | ✓ | | ✓ | |
| [27,70] | Survey | ✓ | ✓ | ✓ | | |
| [72–74,110–112] | Financial Tools | ✓ | | | ✓ | ✓ |
| [71] | Marketing | ✓ | | | ✓ | |
| [75–79,97] | Indicators | ✓ | ✓ | ✓ | ✓ | |
| [31,59,80] | Communication | ✓ | ✓ | | ✓ | ✓ |
| [83,85–87] | LCA | | ✓ | ✓ | ✓ | |
| [20,23,32,38,62,80,88,89] | Materials and Waste | ✓ | ✓ | | ✓ | |
| [48,90] | Risk Management | | ✓ | ✓ | | ✓ |
| [91–93,95] | Training | ✓ | | ✓ | | ✓ |
| [69,92] | Industrial network | ✓ | | ✓ | | |
| [79,82] | Design for disassembly | | ✓ | ✓ | ✓ | ✓ |
| [36,94,100,101] | Equipments | | ✓ | ✓ | ✓ | ✓ |
| [11,41,96] | Technical parameters | ✓ | | | ✓ | ✓ |
| [99] | Plants | ✓ | ✓ | | ✓ | |
| [15,102,103] | Design of the yards | ✓ | ✓ | | ✓ | ✓ |
| [104,105,108] | Holistic Laws | ✓ | ✓ | ✓ | ✓ | ✓ |
| [106,107,113] | Hard Law | ✓ | ✓ | | | ✓ |

A starting point for the implementation of a CE scheme is market analysis. A thorough study of the market situation, attached to the availability of recoverable materials and the demand for materials required by companies, can have the effect of planning a secure market that respects environmental and social criteria, allocating the required materials for industrial developments right away. Enhancing internal and external communication of information through the utilization of a questionnaire can mitigate the information asymmetry among various stakeholders. The questionnaire can also improve the marketing strategies of European shipyards, increasing their competitiveness compared to shipyards located in Southeast Asia. The use of questionnaires also allows the acquisition of data for the construction of robust indicators. The use of both quantitative and qualitative indicators should concern the entire life-cycle of a ship, from design to demolition. The use of such indicators should consider the three dimensions of sustainability to measure their performance. The use of environmental indicators is already consolidated in LCA studies; a wider spectrum measurement is the use of S-LCA indicators for the social sphere. The indicators, in addition to monitoring sustainability themes, should also embrace CE themes, such as the measurability of the circularity of materials.

Based on harmonized indicators within the ship recycling sector, financial companies and public actors can assign incentives to shipping companies and shipyards that comply

with criteria for good environmental and social management, as required by the European taxonomy. Increased funding for employee training in Europe can allow for the attraction of local labor and increase knowledge on waste management, risk management, and waste disposal. The establishment of industrial networks, such as industrial symbiosis, can have significant benefits for the ship recycling sector, including the exchange of information, increased shipyard revenue, reduced costs (procurement, production, waste management, transportation), and access to more favorable tax systems. The design of the shipyard should take into consideration which facilities and equipment should be implemented—for instance, facilities that can reclaim materials and manage waste produced from demolition can bring significant economic benefits to the shipyard and reduce the transportation impacts of such materials. More stringent laws may impede the development of a European market for the exchange of materials from ship demolition. Laws should embrace an incentive/disincentive approach, favoring shipping companies that abide by the ethics of recycling and penalizing those that engage in environment- and human health-damaging behavior. Additionally, regulations should take into consideration the end-of-life recycling phase from the construction phase with a design for disassembly.

To promote the recovery of ship materials, which will encourage the development of a commodity market, a multitude of indicators, incentives, and disincentives can be employed at the national and international levels. For instance, authorities may set ship-specific material recovery targets at the end of their life cycles and offer tax incentives to businesses who achieve those targets. Additionally, it is possible to evaluate the environmental effects of ship dismantling and encourage more environmentally friendly recovery techniques by using environmental indicators such as the carbon footprint. Financial incentives, such as grants or facilities, could also be used as incentives to get businesses to invest in cutting-edge technologies for salvaging ship parts. Moreover, a certification program might be established for businesses that use environmental best practices when dismantling ships, which could improve the company's reputation and market accessibility. On the other side, in order to encourage more sustainable behavior, disincentives, such as taxes or penalties, may also be implemented for businesses that do not adhere to best standards for ship disposal. This will motivate the industry to support the shift to a circular economy and make more ecologically friendly decisions.

The recycling process involves various engineering operations and management activities for demolition and separation. Ship recycling should be described and considered as a modern industrial activity rather than an impactful and dangerous one. The current situation of the high destructiveness of ship demolition activities and the future increase in the number of EoL ships require joint efforts to limit hazardous waste and avoid damage to human health. Circular economy (CE) is gradually being embraced in political and business circles as a promising industrial environment that can leverage multiple benefits while saving resources and promoting sustainable growth and social inclusion. The recommendation is to move from a linear economy (based on production, use, and waste) to a model that prioritizes the reuse of equipment, materials, and components, minimizing waste at the end of the process.

The article is introduced as a thorough and in-depth examination of the condition of ship recycling at the moment, demonstrating the existing landscape of the industry and providing a methodical analysis that ensures the implementation of a circular economy model. While the environmental and social aspects are analyzed and combined in a well-defined region of South-East Asia, the literature on ship recycling only marginally considers the argument of the EC as a potential winning strategy, limiting normative measures and the performance of certification. Currently, there exist technologies and solutions that could help the industry realize and create a circular business model, but these must be evaluated and used in accordance with a correlation between the previously established criteria. In order to increase recycling in Europe, which is essential for increasing competitiveness and raising the safety and environmental standards of shipyards in South-East Asia, European policy actors must involve funding measures, such as incentives and subsidies. This

is because the realization of an EC strategy can only be achieved through a top-down approach. Only thorough dialogue between all stakeholders on the shared issue of resource value can result in the creation of a market. To be able to create a secure new business for the maritime industry, it is also vital to evaluate their true economic, financial, and environmental viability.

The introduction of the theoretical conceptual framework necessary for the transition to the circular economy and the possibility of creating a circular economy model are offered.

*5.1. Economic Proposal*

To address the challenges and issues facing ship recycling in Europe, there are several potential solutions that can be explored. One important step is to conduct more comprehensive market studies, focusing on the materials salvaged and the environmental impacts of the demolition process in different regions. By adopting a cradle-to-grave perspective, researchers can gain a better understanding of the full life cycle of a ship and the impact of its recycling on the environment. Another proposal is to conduct more LCA studies that can provide a comprehensive comparison of different recycling methods. This will help to identify the most effective and environmentally friendly techniques, while also highlighting areas where improvements can be made. Reliable economic, environmental, and social indicators are also essential to ensure that the recycling process is as safe and sustainable as possible, but they must especially consider the circularity of the materials and specialized equipment needed to manage hazardous substance spills and protect workers from potential harm. By investing in the development and deployment of this equipment, ship recycling companies can reduce the risk of environmental damage and protect the health and safety of their employees. In addition, it is important to examine the whole life cycle and recycling procedures used worldwide. By doing so, researchers can identify best practices and lessons learned that can be applied to the European market. Finally, there is a need to extend the focus of economic research on ship recycling and improve understanding of the effects and hazards associated with the process. This will allow for the development of more effective efforts.

*5.2. Environmental Proposal*

One important step towards sustainable ship recycling is the use of reliable environmental indicators throughout the ship recycling process. This can help to ensure that the process is conducted in an environmentally responsible manner, minimizing the negative impact on ecosystems and communities in the surrounding area. Another key solution is the inclusion of a comprehensive comparison of recycling methods used in shipyards. This can help to identify the most effective and environmentally friendly methods for breaking down and recycling ships and ensure that best practices are followed across the industry. In addition to examining the methods used during the demolition phase, it is important to examine the whole life cycle of ships. This can help to identify opportunities for reducing the environmental impact of ships at every stage of their existence, from design and construction to use and disposal. A more thorough analysis of recycling procedures used in shipyards worldwide is also needed. This can help to identify areas where improvements can be made and ensure that best practices are followed across the industry. Furthermore, developing specialized equipment for managing hazardous substance spills during ship breakdown and recycling is essential for ensuring the safety of workers and minimizing environmental damage. Extending the focus of environmental research on ship demolition is also critical. This can help to identify new technologies and methods for recycling ships in a safe and sustainable way. Finally, effective efforts to mitigate environmental effects and hazards associated with ship demolition are needed. This can include measures such as the use of protective equipment, safe handling procedures for hazardous materials, and the development of policies and regulations that prioritize sustainability and safety.

### 5.3. Social Proposal

To address this issue, several solutions can be implemented. It is crucial to evaluate the working conditions and morals in South-East Asian yards to ensure that workers are not subjected to dangerous or unethical practices. This can be achieved through regular monitoring and audits of the yards by independent organizations. Encouraging higher standards of morality and safety in South-East Asian shipyards is crucial to ensuring that ship recycling practices are sustainable and ethical. This can be achieved through the development of industry standards and the promotion of best practices by stakeholders. The social sphere should be included in conventional life cycle assessment (S-LCA) techniques with pertinent indicators. This will provide a more comprehensive understanding of the impact of ship recycling on society and help identify areas where improvements can be made. Increasing knowledge on work management and organization in Europe can help to improve the efficiency and safety of ship recycling practices. This can be achieved through the establishment of training programs and industrial networks that promote best practices and collaboration among stakeholders. Expanding the workforce in Europe can also improve the working environment and safety in South-East Asian shipyards. This can be done by providing opportunities for European workers to participate in ship recycling activities in South-East Asia, thereby sharing knowledge and expertise.

### 5.4. Technology Proposal

One proposal is to apply ship demolition technologies that can be applied in other shipyards around the world. This approach can facilitate the adoption of best practices and the transfer of knowledge, leading to greater efficiency and more standardized practices in ship recycling. Moreover, it is essential to involve the whole ship sector supply chain in the recycling process, from shipbuilders to shipowners and operators, and downstream to the recycling facilities. This will help to ensure that everyone involved is committed to sustainable practices and shares the responsibility for reducing the environmental impact of ship recycling. To reduce the environmental effects of ship recycling, a risk management approach should be adopted which includes hazardous material conversion and leakage monitoring. Such measures can help to prevent contamination and protect the health of workers and the environment. It is also important to implement a circular economy approach to ship recycling, which involves reusing and repurposing materials and products, as well as reducing waste and emissions. To achieve this, the whole ship sector supply chain should be involved in the implementation of circular economy practices. Finally, it is crucial to assess the economic and strategic viability of ship recycling, considering each nation's recovery chain. This will help to ensure that ship recycling is sustainable in the long term and that it benefits both the economy and the environment. Furthermore, the use of information technology and communication (ITC) can enhance the efficacy and reproducibility of ship recycling practices, while also improving the monitoring of environmental effects and ensuring compliance with regulations.

### 5.5. Regulatory Proposal

The issue of ship recycling in Europe is a complex one that requires a multifaceted approach to ensure sustainable and responsible practices. One of the first steps to achieving this is to clearly define the terminology surrounding ship waste. By doing so, we can better understand and address the challenges associated with ship recycling. Enacting stricter legislation is another crucial aspect of improving ship recycling practices in Europe. By implementing more robust regulations, we can hold companies accountable for their actions and ensure that they are operating in an environmentally responsible manner. In addition, increasing stakeholder collaboration is essential to creating a sustainable solution for ship recycling. This includes working with ship owners, shipyards, and other industry stakeholders to develop effective strategies for managing ship waste and reducing environmental impact. Adopting a holistic approach is also critical to achieving sustainable ship recycling practices. This means considering the entire life-cycle of a

ship, from its design and construction to its use and eventual disposal. Encouraging active participation among all value chain actors is another important step in improving ship recycling in Europe. This requires educating and engaging industry stakeholders, such as ship owners, operators, and regulators, to ensure that they understand their roles and responsibilities in creating a sustainable and responsible ship recycling industry. Establishing top-down political and regulatory frameworks is also essential to creating a sustainable ship recycling industry. This includes developing policies and regulations that support environmentally responsible practices and hold companies accountable for their actions. Implementing financial tools, such as insurance, port fees, and licensing, can also help to promote sustainable ship recycling practices. These tools can encourage companies to invest in sustainable practices by offering financial incentives for compliance. Using certification procedures is another important aspect of creating a sustainable ship recycling industry. Certification programs can help to ensure that ship recycling practices meet established environmental standards and best practices.

### 5.6. Interconnection with the Proposals

Through the study of the examined clusters and the search for valid variables for the transition to a circular economy model for the ship recycling sector, a Venn Graphic was developed according to the indications of [115]. The Venn diagram is an effective tool for displaying intersections between sets. This gives one a comprehensive perspective of the system and enables one to see where the suggested improvements converge in order to move the model closer to the circular economy. The Venn Graphic aims to provide both a holistic view of ship recycling and to direct future research towards new empirical studies for the sector that can promote a new circular market for resources, adapting them to circular economy strategies. The Figure 4 represented Venn Graphic.

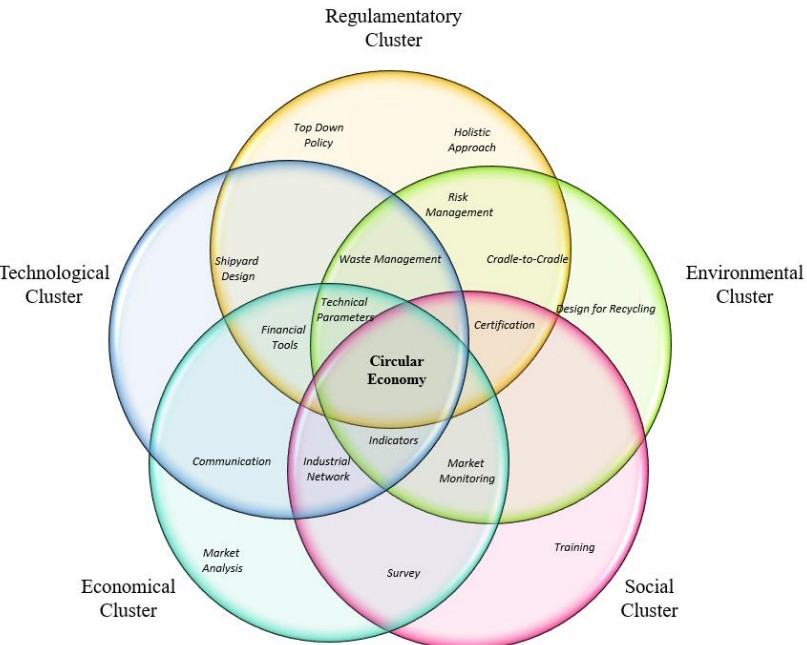

**Figure 4.** Venn Diagram Source: authors' elaboration.

Efforts are needed to develop successful business strategies for the sector's shift towards a circular economy (CE). Improving dialogue and communication between all parties and establishing coordination mechanisms are necessary for ship recycling. Financial incentives and disincentives can drive the adoption of better, more environmentally safe technologies for workers. This can encourage ship owners to scrap their ships safely and reduce the use of flags of convenience. Worker training, especially in Europe, should also be incentivized.

Indicators can improve management and planning during the demolition phase and should be considered during the ship design phase to measure circularity performance.

Using design for recycling and the cradle-to-cradle method for recycling ships may have positive effects on the economy, society, and environment. Moreover, by fostering the growth of new knowledge and technologies in Europe, the use of these techniques can lower the costs associated with ships' ultimate disposal, boost industry profitability, and contribute to the creation of new employment opportunities in the ship recycling industry. It can also increase worker safety and lessen the environmental effect of South-East Asian ship recycling and dismantling operations. However, for these benefits to be realized, both public and private actors must work together. The use of design for recycling and a cradle-to-cradle strategy in the ship recycling industry must be encouraged by government agencies while upholding environmental regulations and workers' rights. In order to use these strategies and increase their competitiveness, businesses must engage in the research and development of the technologies and competencies necessary. In Europe, using tools such as surveys that are given to shipyards and shipowners can increase dialogue among the stakeholders and the environmentally responsible management of debris created during ship disassembly. This can assist in enhancing the economic, social, and environmental character of the industry. Additionally, waste management must be strict and in compliance with European rules to maintain respect for the environment and public health. Establishing approved standards may be a useful method to improve demolition's sustainability and ensure correct waste disposal. It is also crucial to take into account the technical parameters of the ship recycling facilities, which need to operate efficiently and in accordance with environmental laws. In addition, it is essential to evaluate the environmental, social, and economic effects of the demolition process by measuring all aspects of sustainability using reliable indicators. Finally, the adoption of the principles of the circular economy represents one of the main challenges for the field of the ship recycling. This implies rethinking the entire life cycle of ships, from design to demolition, in order to maximize the recovery of materials and minimize their impact on the environment.

## 6. Conclusions

Hypothesizing a green market for the ship recycling sector translates into a market where the optimization of the secondary raw materials flow management, according to an "end of waste" approach, encourages the development of a business-to-business trade for the supply of secondary raw materials and critical raw materials to improve upstream design and downstream recovery for new industrial processes. It must be considered that future developments are heavily conditioned by the different variations in the price of the materials contained in ships. The economic value of the materials of a ship at its end of life can be the source for the creation of secondary markets of critical raw materials. From an eco-neutral perspective, a green ship has all its environmental aspects under control. It would be the result of circular design, designed with advanced methods of material and energy use, in which all components are designed to be continually reused in future projects, reducing waste.

The article proposed an in-depth analysis of the state of the art of ship recycling, analyzing the different clusters that match the applications of the EC. Scarcity in the implementation of circular economy in the analyzed clusters is recognized. The prospect of creating a circular business model for recycling ships has not yet been completely addressed in the literature, and many of the studies are limited to a small geographic region. There are very few studies from Europe that address the issue. Nonetheless, the economic, technological, and judicial resources required to create a circular business model for ship recycling already exist. The stakeholders involved, including the governments, shipowners, shipyards, and non-governmental organizations, must work together to achieve this aim. In order to transition the business of ship demolition towards a circular model, potential options for implementation in the various clusters under consideration are suggested in Figure 4. Additionally, to create a circular market, it would also be helpful to know when

the ships will become waste. It is likely that a trial of a circular economy strategy for European shipyards and European shipowners has not yet been possible, but elaborate proposals can affect the event of a circular business model of resources for the development of new circular economic strategies for ship recycling, when comparing and applying the different variables discussed in the discussion phase and shown in Table 2. Empirical and experimental research are required to expand the field of study.

To address the research questions posed, the shipping recycling sector is well-prepared to find solutions incorporating circular economy principles and to implement circular business models. To achieve it, the sector must consider the demand for ship components, fleet age, and adverse conditions. Integrated management systems, material flow analysis, lack of information asymmetry, faster communication based on ITC systems, improved quality certification, material flow mapping, and more flexible and less restrictive regulations can create a more efficient and circular market. As the global merchant fleet is expected to grow, ship dismantling is expected to increase, requiring solutions that align with sustainability and circular economy principles. Europe aims to make the sector more ecological and competitive through the Circular Economy Action Plan and European Green Deal priorities.

The article is distinctive in how thoroughly it connects circular economy ideas with ship recycling and life-cycle management practices. This paper attempts to present a comprehensive assessment of the problems and opportunities connected with the application of circular economy techniques in the ship value chain, whereas prior literature has concentrated on certain areas of sustainability or research clusters. The authors have identified critical areas for the possible development of circular economy models for ship eco-design, life cycle management, and recycling by examining and categorizing scientific papers. The novelty of this article rests in its emphasis on the development of green markets and the potential for cutting-edge economic models in the context of ship recycling in the EU, in addition to their environmental advantages. Overall, this research contributes to a more complete understanding of the potential for circular economy principles to drive sustainable practices in the ship recycling industry.

Future research paths should concentrate on empirical study with an interdisciplinary and holistic approach to the area in order to advance towards the opening and development of a business model based on the principles of the circular economy for the ship recycling industry. To gather precise and verified data, empirical field research is crucial. Direct observations, interviews, and analysis of recycling activities at specific locations should be part of field research. It will be crucial to collect both quantitative and qualitative data, for instance, on the quantities and types of materials recovered, on worker safety procedures, their health and working conditions, and on the environmental and social impacts. Another important factor is the participation of all stakeholders, including authorities, shipowners, enterprise, and shipyards. Understanding the needs and perspectives of all parties involved might help to find potential solutions. The development of best practices for ship recycling operations should be based on the results of empirical study.

This article has limitations, including that the recruitment of articles was done without considering secondary search engines and that conference or book samples were not included in the active sample. Moreover, literature studies might only analyze the situation in a specific geographic area, ignoring the cultural, legal, and economic variations that affect ship recycling in other areas of the world.

**Supplementary Materials:** The following supporting information can be downloaded at: https://www.mdpi.com/article/10.3390/su15075919/s1. PRISMA Checklist.

**Author Contributions:** Conceptualization, F.T.; Methodology, F.T.; Validation, E.M.M.; Data curation, M.M.; Writing—original draft, M.G.; Writing—review & editing, E.M.M.; Supervision, M.M. All authors have read and agreed to the published version of the manuscript.

**Funding:** This research received no external funding.

**Data Availability Statement:** The authors agree on the set of articles used for the analysis of literature after information by email. Contact us if you need the article database.

**Conflicts of Interest:** The authors declare no conflict of interest.

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
