# Peer review of "Perspectives for the Development of a Circular Economy Model to Promote Ship Recycling Practices in the European Context: A Systemic Literature Review"

_sustainability, doi:10.3390/su15075919_

Round 1

Reviewer 1 Report

1. This paper is a review article with aims to enhance the current state of ship recycling with the circular economy. From the perspective of engineering and techno-analysis, the objective of this paper is weak. It suggests elaborating more to find the correlation between ship recycling's impact on a circular economy.

2. The author states the goal is also to develop a conceptual framework for the effective recirculation of components and raw materials for ship recycling. The conceptual framework did not found in the result and discussion, the author should seriously answer the goal by adding a more comprehensive result of the conceptual framework of ship recycling.

3. Figure 1 is less information, it should add geographical information (lat, long), name of the country, etc.

4. The result has not been deeply analyzed, most of the content is only a description from other research. The author should analysis what is the relation between the result of another researcher. Add quantitative and qualitative analysis.

5. The conclusion should be concise and answer the objective and the goal of the study.

Author Response

Dear reviewer,
thank you for your suggestions and for appreciating our efforts. Thank you again for your time and appropriate requests for revision of the article. The manuscript has been revised according to your most appropriate requests.
-We enriched the explanation about the correlation between ship recycling's impact on a circular economy into the result section, discussion section and conclusions. 
-We better highlighted the explanation of the goal of the work. In this regard, a deep analysis has been added for the introductive conceptual framework also by adding a new table within the discussion section.
-We modified Fig.1 according to your suggestions
-We revised and integrated, more clearly, the result section. Research findings shown in the paper have been deeply addressed in light of the literature analysis.
-The conclusion has also been revised in light of greater clarity with regard to the link between the objective and the goal of the study.

Reviewer 2 Report

The topic of the article and the research problem raised is current and important. The issue of recycling scrapped ships is a serious environmental problem, and especially in South Asian countries, where more than 90% of ships are handed over for scrapping.

The structure of the article is not objectionable. It is written on the basis of a rich review of the literature and previous scientific reports in this area. 

The authors of the article have made a thorough review of the literature, but also international normative acts concerning the process of design, construction and recycling of ships, including above all the issue of inventory of hazardous materials from the scrapping of ships after their decommissioning.

The authors clearly and comprehensibly described the methodology of the qualitative research used, indicating the stages of the research process and selection criteria. However, the conclusions lack a reference to the main purpose of the article, questions and research hypotheses. In addition, I also did not find a description of the limitations of the conducted analyzes and recommendations for future research.

Technical notes

The authors should indicate the source of information or data used, or the author under each table and figure. This should be completed.

Author Response

Dear reviewer,
thank you for your suggestions and for appreciating our efforts. We thank you again for your time and appropriate requests for revision of the article. The manuscript has been revised according to your most appropriate requests.
-Conclusion has been revised in the light of greater clarity concerning the link between the objective research questions and the goal of the study. Limitations for the study have been added within the conclusion section. 
- Technical notes and references have been revised.

Reviewer 3 Report

The article attempts to develop a conceptual framework for the effective recirculation of components and raw materials through literature review and categorization into strategic clusters to establish a path for possible developments of a circular economy model that would support eco-design, waste management ship lifecycle, and recycling.

It is an interesting and current topic and the article is concise, clear, and well-written.

However, it would improve if the authors would hypothesize and expand the discussion section. In addition, the conclusion also seems very short and generic, leaving room for more consideration on the subject, and does not explicitly refer to the answer to the research questions.

References must be formatted in accordance with the journal's rules. 

Author Response

Dear reviewer,

thank you for your suggestions and for appreciating our efforts. We thank you again for your time and appropriate requests for revision of the article. The manuscript has been revised according to your most appropriate requests.

-We revised and integrated more clearly the Discussion section. Research findings shown in the paper have been deeply addressed in light of the literature analysis.

-Conclusion has been revised in the light of greater clarity with regard to the link between the objective research questions and the goal of the study. 

Round 2

Reviewer 1 Report

The revision is not clear.

The author ignores constructive comments from reviewers.

Author Response

Dear reviewers

thank you for your suggestions and for appreciating our efforts. We thank again you for your time and appropriate requests for revision of the article. The manuscript has been revised according to your most appropriate requests.

We reviewed the article with your information. We have better presented the conceptual framework and proposed a graph for the interconnections of the proposals. We have better explained the gaps in research and the results obtained.